# PRESERVING SEMANTICS IN TEXTUAL ADVERSARIAL ATTACKS

## ABSTRACT

Adversarial attacks in NLP challenge the way we look at language models. The goal of this kind of adversarial attack is to modify the input text to fool a classifier while maintaining the original meaning of the text. Although most existing adversarial attacks claim to fulfill the constraint of semantics preservation, careful scrutiny shows otherwise. We show that the problem lies in the text encoders used to determine the similarity of adversarial examples, specifically in the way they are trained. Unsupervised training methods make these encoders more susceptible to problems with antonym recognition. To overcome this, we introduce a simple, fully supervised sentence embedding technique called Semantics-Preserving-Encoder (SPE). The results show that our solution minimizes the variation in the meaning of the adversarial examples generated. It also significantly improves the overall quality of adversarial examples, as confirmed by human evaluators. Furthermore, it can be used as a component in any existing attack to speed up its execution while maintaining similar attack success [1].

## 1 INTRODUCTION

Deep learning models have achieved tremendous success in the NLP domain in the past decade. They are applied in diverse critical areas such as hate speech filtering, moderating online discussions, or fake news detection. Successful attacks on these models could potentially have a devastating impact. In recent years, many researchers have highlighted that language models are not as robust as previously thought (Jin et al., 2020; He et al., 2016; Zhao et al., 2018; Szegedy et al., 2014; Kurakin et al., 2016a;b) and that they can be fooled quite easily with so-called adversarial examples, which introduce a small perturbation to the input data 'imperceptible' to the human eye. For example, in the domain of offensive language detection, we can have an offensive text on input and modify it in such a way that the meaning is preserved, but the modified text will fool the system to classify it as non-offensive (Jin et al., 2020). A similar scenario is illustrated in Figure 1 in the domain of sentiment analysis of movie reviews.

Although adversarial examples can be perceived as a threat, they also help us identify and understand potential weaknesses in language models and therefore contribute to the defense system, threat prevention, and decision making of the models just as well (Ribeiro et al., 2018). Furthermore, when attacks are included in training data, the general robustness of the model and its ability to generalize can be improved (Goodfellow et al., 2014; Zhao et al., 2018).

Regarding the imperceptibility of adversarial attacks, it is easily definable in a continuous space, in domains like audio or vision. In computer vision, imperceptibility is a certain pixel distance between the original image and its perturbed version (Chakraborty et al., 2021). However, this term is much more difficult to grasp in discrete domains such as text, where there is no clear analogy and where an indistinguishable modification simply cannot exist. This is why several definitions of a successful adversarial example have been developed that are specific to discrete domains such as text (Zhang et al., 2020b).

According to (Jin et al., 2020) we can identify three main requirements for an adversarial attack on text to be successful:

---

[1]The code, datasets and test examples are available at `https://github.com/`

1. Human Prediction Consistency: The prediction made by humans should remain unchanged.

2. Semantic similarity: The designed example should have the same meaning as the source as judged by humans.

3. Language fluency: The generated examples should look natural and grammatical.

We can observe a similar constraint definition in (Morris et al., 2020a), with the addition of the "overlap" constraint that focuses on character-level changes.

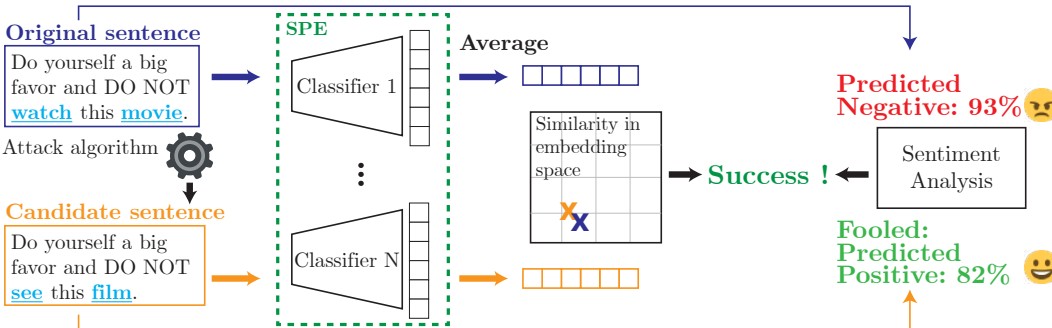

Figure 1: The typical adversarial example pipeline for a sentiment classification task. The attacked classifier was trained to distinguish between positive and negative movie reviews. The semantic similarity metric is based on our Semantics-Preserving-Encoder (SPE), which embeds sentences using an array of supervised classifiers. In this example, substituting two words simultaneously fools the model into changing its prediction and passes the semantic similarity test.

Although most adversarial attacks claim to meet these constraints (Gao et al., 2018; Li et al., 2021b; Garg & Ramakrishnan, 2020), careful scrutiny shows otherwise. We observed that many adversarial examples do not preserve the meaning of the text in some cases. This is also supported by (Morris et al., 2020a) whose findings are similar. To overcome this problem, (Morris et al., 2020a) suggested increasing cosine thresholds and introducing mechanisms such as grammar checks to improve the quality of adversarial examples. However, it is at the cost of the attack success rate, which decreased rapidly by more than 70%. We suggest a different solution that promises to avoid this decline in the attack success rate.

It appears that the problem lies in the similarity metric itself, whose function is to measure the difference between the original and perturbed sentences. These metrics mostly use encoders that are trained with limited supervision. This makes them more susceptible to problems with antonym recognition. Because the antonyms are used in a similar context in the training data, the encoder assumes that they are alike. As a result, sentences such as *'This movie is so good'* and *'This movie is so bad'* are considered similar in this case, as illustrated in the third column of Table 1.

Building on these findings, we propose a new sentence encoder for similarity metrics in textual adversarial attacks called Semantics-Preserving-Encoder (SPE). SPE is trained with full supervision on annotated datasets. Thus, it should be more robust towards the antonym recognition problems that we observed frequently in adversarial examples.

These premises were proven to be valid throughout the experiments, where we compared our solution to some of the most common similarity metrics used in adversarial attacks. The results show that our solution largely reduces the occurrence of text meaning modification and also significantly improves the overall quality of the adversarial examples generated, as confirmed by human evaluators. Furthermore, our solution – SPE can be integrated into any existing adversarial attack as a component granting a much faster execution and comparable attack success rate.

In summary, we consider our main contributions to be as follows:

1. We propose a simple, but powerful sentence encoder – SPE, which improves the overall quality of adversarial examples (minimizes text meaning modification and antonym recognition problem). SPE can also be used as a component in any existing attack to speed its execution while maintaining a similar attack success rate.

2. We propose a new metric for evaluating the quality of the adversarial examples – rASR, which reflects the real performance of adversarial attacks.

3. We evaluate some of the most common sentence encoders used in adversarial attacks both manually and automatically on relevant datasets such as hate-speech and offensive language detection (Barbieri et al., 2020), in addition to other popular classification tasks such as Yelp Reviews (Zhang et al., 2015) and Rotten Tomatoes (Pang & Lee, 2005).

4. We release our work as open source, including the code, human evaluations, datasets, and test samples for the purpose of reproducibility and future benchmarking.

## 2 RELATED WORK

**Textual adversarial attacks.** There were many attempts to create textual adversarial attacks that preserve semantics and are grammatically correct. Ultimately, we can distinguish between three different ways in which other researchers approach this. The first approach aims to modify the whole sentence using various sophisticated phrase perturbations, such as paraphrasing (Gan & Ng, 2019; Wang et al., 2020). However, these modifications often have problems with semantic preservation (Zhang et al., 2020b).

The second approach focuses on character-level modification, such as misspellings or typos, which has proven to be more successful in terms of semantics preservation (He et al., 2021; Li et al., 2019; Ebrahimi et al., 2018). However, research shows that these types of attacks can be mitigated quite easily with tools such as grammar checks (Pruthi et al., 2019; Jones et al., 2020).

Lastly, the word-level attack technique focuses on the substitution or modification of a single word (or combination of multiple words) in the text (Alzantot et al., 2018; Ren et al., 2019; Dong et al., 2021; Li et al., 2021a). This type of attack aims to preserve the constraints defined by (Morris et al., 2020b), often using methods such as synonym substitution to improve semantic preservation.

**Similarity metrics.** The majority of word-level adversarial attacks enforce semantics similarity by using Universal-Sentence-Encoder (USE) (Cer et al., 2018) or BERTScore (Zhang et al., 2020a) encoders, both of which are trained mainly on unsupervised tasks. USE is trained on a task such as Skip-Thought (Kiros et al., 2015), a conversational input-response task (Henderson et al., 2017) and a supervised classification task performed on the SNLI dataset (Bowman et al., 2015). BERTScore is based on a pre-trained BERT language model (Devlin et al., 2018), which was also trained unsupervisedly on the Next Sentence Prediction and Masked LM tasks.

Even though these sentence encoders have been thoroughly studied on various general tasks, only a few previous works acknowledge their flaws when used in adversarial attacks (Morris et al., 2020b; Herel, 2022). In most cases, these encoders struggle to recognize changes in the text's meaning and semantics. To overcome this, (Morris et al., 2020b) increased the cosine threshold, resulting in an improved quality of adversarial examples, but over 70% decline in the attack success rate.

Looking at the big picture of the observed problems, we have discovered a potential link to the encoder training process. We show that the unsupervised training predetermines these encoders to have problems with antonym recognition, which leads us to introduce our own Semantics-Preserving-Encoder.

## 3 METHOD

In order to fully grasp the idea and motivation behind our sentence encoder – Semantics-Preserving-Encoder, we formulate the textual adversarial attack problem in the following section. Next, we formally introduce SPE, together with its classifiers and other properties.

### 3.1 PROBLEM FORMULATION

Most similarity metrics in adversarial attacks rely on sentence encoders such as USE (Cer et al., 2018) or BERTScore (Zhang et al., 2020a). These encoders use multi-task learning with a high emphasis on unsupervised tasks. The BERT language model (Devlin et al., 2018) uses Masked-Language Modeling. USE is trained on Skip-Thought (Kiros et al., 2015) like task, where the goal

is to predict the middle sentence based on the given context. Both of these training methods could be seen as a variation of a Skip-Gram/CBOW model (Mikolov et al., 2013), where the goal is to predict the context of a given target word, and vice versa for CBOW. However, this training method forces synonyms to be mapped into a similar vector space as antonyms, as they appear in the same context. Therefore, two contradictory sentences in their vector representation could be very close to each other in the vector space despite their opposite meaning. That is if the unsupervised training with Skip-Gram or CBOW like tasks is used. This is the case for both BERT and USE.

A typical example of this problem is shown in Table 1. In this example, we used USE to encode the sentences into the vector space and then measured the cosine similarity. The major flaw of the encoder can be observed immediately. Sentences with totally opposite meanings, *"This movie is so good"* and *"This movie is so bad"*, are closer to each other than sentences with almost the same meaning, *"This movie is so good"* and *"This movie is so tasty"*.

## 3.2 SEMANTICS-PRESERVING-ENCODER

The core idea of our encoder lies in supervised training. We took advantage of the existing pre-labeled datasets and utilized them in the training data to tackle the problem with opposite words appearing in the same context. As a result, the words that are the most discriminative for the given label will be close to each other in the vector space. Thus, the sentences mentioned above *'This movie is so good'* and *'This movie is so bad'* should never be close to each other in the vector space, because their semantics label will be exactly opposite.

We have combined multiple classifiers trained on different annotated datasets, which allows us to have a diverse set of different sentence vectors. The sentence vectors will differ because each classifier produces its vector according to the task on which it was trained. Therefore, the diversity of classifiers implies a diverse set of sentence vectors. Moreover, by combining several sentence embeddings from different classifiers, we can create a robust classifier that can produce a high-quality embedding for a broad range of topics.

From a sentence $S$, an attack will generate a candidate adversarial example $S^*$. We denote the $N$ supervised classifiers by $C_1, C_2, \ldots, C_N$. For simplicity, we consider these classifiers to be functions whose outputs are the sentence embeddings extracted from the classifier when applied to a sentence. Formally, for all $k$, $C_k(S) = \boldsymbol{e}_k \in \mathbb{R}^p$ where $p$ is the embedding dimension. The complete embedding of a sentence is obtained by averaging the output of the classifiers into a single embedding vector.

$$\boldsymbol{e}^{(S)} = \frac{1}{N} \sum_{k=1}^{N} C_k(s) = \frac{1}{N} \sum_{k=1}^{N} \boldsymbol{e}_k^{(S)}.$$

The similarity between the original sentence $S$ and the attacked sentence $S^*$ is computed with the cosine distance between their embeddings as follows

$$\text{Sim}(S, S^*) = \frac{\boldsymbol{e}^{(S)} \cdot \boldsymbol{e}^{(S^*)}}{\|\boldsymbol{e}^{(S)}\| \|\boldsymbol{e}^{(S^*)}\|},$$

where $\cdot$ represents the dot product between vectors in $\mathbb{R}^p$ and $\| \cdot \|$ is the $L_2$ norm in $\mathbb{R}^p$. For a threshold $\epsilon$, $S$ and $S^*$ will be considered to have the same meaning if $\text{Sim}(S, S^*) > \epsilon$.

The classification model that we used is fastText (Joulin et al., 2016). However, it is important to note that any other classification model can be used instead. We decided to use fastText due to its many advantages. Firstly, fastText classifiers allow us to create sentence vectors rather quickly with a reasonable performance for the given task. Secondly, we can reduce the dimensionality of the vector space. This way we can put more information into fewer dimensions, which results in a more efficient space storage.

The classifier selection and training process are key to achieving a robust solution with high-quality results. We begin to train classifiers on Natural Language Inference (NLI) datasets like SNLI (Bowman et al., 2015), cola, rte and sst2 (Wang et al., 2018), to help SPE eliminate the problem with out of vocabulary words. We have further extended this selection with classifiers trained on more downstream tasks such as emotion classification CARER, Yelp reviews and Stack Overflow questions classification (Saravia et al., 2018; Zhang et al., 2015; Annamoradnejad et al., 2022). This should help our approach to map words with a different meaning to different vectors. However, it is

possible that for other purposes than adversarial attacks, a more optimal set of classifiers could be found. We then took the sentence embeddings from each classifier and averaged them into one. This process is illustrated in Figure 2. More details about the selection of classifiers and the averaging method can be seen in Appendix A.1.

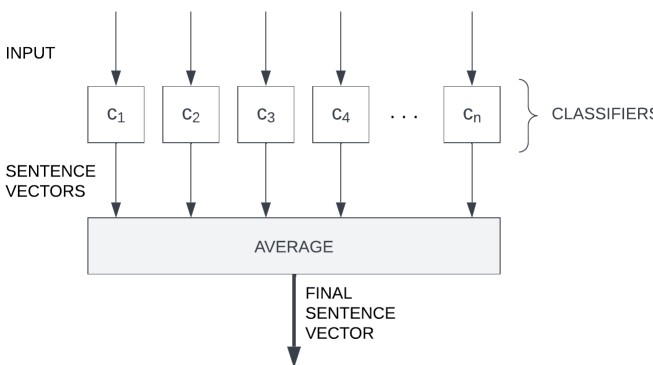

Figure 2: Model architecture for SPE. For the given input, each classifier creates the sentence embedding. These sentence vectors with the same dimension are averaged to produce the final sentence vector.

Furthermore, we show that our approach can better distinguish between a text with opposite and similar meanings than others that use limited supervision. This is illustrated on the previously remarked example in Table 1, where SPE achieves a much larger difference in cosine similarity, which is much more indicative of the real meaning of the sentence.

| Original sentence | Perturbed sentence | Cosine similarity (USE) | Cosine similarity (SPE, ours) |
|---|---|---|---|
| *This movie is so good* | *This movie is so **bad*** | 0.90 | **0.63** |
| *This movie is so good* | *This movie is so **tasty*** | 0.89 | **0.95** |

Table 1: Our approach, SPE, is able to better distinguish between the sentences with similar and opposite meaning than sentence encoders trained with mostly unsupervised methods like USE (Cer et al., 2018). Texts in the table were transformed into latent space by sentence encoders and then the cosine similarity was measured. Sentences with an almost identical meaning should have a higher cosine similarity than sentences with an opposite meaning. This is the case for our approach – SPE.

As mentioned above, the use of fastText classifiers allows us to achieve a reduced real-time complexity of SPE compared to USE (Cer et al., 2018) or BERTScore (Zhang et al., 2020a). SPE only needs to perform 7 matrix multiplication operations (because we implement 7 fastText classifiers) with small matrices due to the classifiers hidden layer dimension equal to 10. This is a marginal difference compared to the computations executed by USE (Cer et al., 2018), which performs the operations with 512-dimensional vectors and, therefore, very large matrices. The final time complexity to compute the sentence representation using SPE is defined as follows:

$$O = \sum_{i=1}^{7} V * H_i \tag{1}$$

where $V$ is the length of the sentence and $H_i$ is the size of the hidden layer for i-th classifier.

As the next logical step after the implementation of SPE, we wanted to apply our text encoder in similarity metrics. Generally, the role of an encoder in a similarity metric is to transform the given text into a corresponding vector in the latent space, which is then used to evaluate the similarity. Specifically, cosine similarity is measured between the vectors of the original text and the perturbed

text. If the cosine similarity reaches a certain preset threshold $\epsilon$, an adversarial example is considered successful.

Therefore, for our metric, the general cosine threshold also had to be defined. Ideally, it should be set so that consistent results for any domain are produced. Our experiments empirically show that setting $\epsilon$ to **0.95** performs consistently well across different datasets, which is the reason we chose it for our similarity metrics.

## 4 EXPERIMENTS

To evaluate our proposed sentence encoder in real attack use cases, we performed some automatic and human-based manual evaluation. As explained in Section 1, the commonly used attack success rate is fundamentally flawed and is not enough to measure the success of an adversarial attack. The metric only gives a partial view of the real capability of an attack, obscuring the quality of the semantic similarity constraint on the generated sentences. For this reason, we conducted an extensive survey to collect human evaluations of the quality of the attacks generated, which allows us to evaluate the semantic similarity constraint.

The main goal of our experiments is to study the impact of SPE when used as a semantic similarity constraint in adversarial attacks. If this constraint is too strict, only a few high-quality sentences would be accepted as successful attacks, potentially missing many good candidates. Inversely, too loose a constraint would produce many low-quality examples. We focus our experiments on two widely used attacks: TextFooler (Jin et al., 2020) and its improved version TFAdjusted (Morris et al., 2020b). To understand the effect of SPE in existing attacks, we use it as a constraint, together with two other state-of-the-art semantic similarity metrics, the Universal Sentence Encoder (USE) (Cer et al., 2018) and BERTScore (Zhang et al., 2020a), totaling six attack/sentence encoder pairs.

### 4.1 AUTOMATIC EVALUATION

We automatically evaluate adversarial attacks using three metrics:

**Attack success rate (ASR).** The percentage of successful adversarial examples found by an attack. Given a sentence classifier, a successful adversarial example means that the attack generated a sentence similar to the original that is assigned a different label by this classifier. A higher success rate means that more generated sentences are assigned a different label.

**Time.** Average time needed to create an adversarial example. Attacks may rely on various search techniques to generate candidate sentences, leading to varied generation times.

**Modification rate.** The percentage of modified words. To be as imperceptible as possible, attacks should use as few edits as possible.

### 4.2 HUMAN EVALUATION

Using automatic evaluation, such as the attack success rate, adversarial attacks are considered successful if they simultaneously pass a semantic similarity threshold and manage to change the label of a classifier. Measurement of semantic similarity between sentences is still an open research problem with no commonly accepted solution. In some cases, even humans disagree on whether two sentences are similar or not. For this reason, we include an extra evaluation step in our experiments to obtain more robust results. Sentence pairs were presented to 5 annotators who assigned them an integer score between 1 and 4, where score 1 means strongly disagree, 2 disagree, 3 agree and 4 strongly agree. The higher the score, the more similar the meaning of a pair of sentences will be according to the annotator. Each annotator was given the sentences to label in several online forms. We averaged the scores of all the annotators into a single value for each sentence. Sentences were assigned a binary label; those with an average score greater than or equal to 2.5 are considered similar, while the others are labeled as not similar. Because the original score system is not binary, the threshold was chosen exactly in the middle of these values. Experimentally, we have determined that the exclusion of the equality to 2.5 does not have any noticeable impact on the results.

Based on this evaluation, we estimate a realistic success rate of the attack — or an estimated "real ASR" (rASR). This number represents the number of successful attacks that would actually fool the

reader into thinking that they have the same meaning as the original sentence but have a different label assigned by the classifier. This score, although costly to estimate, gives a very accurate measure of the quality of an attack. A perfect attack would reach a high ASR and a very similar rASR (ASR ≈ rASR), which means that it successfully fooled the target classifier repeatedly and that all the corresponding attacked sentences also conserve the same meaning as the original ones in the eyes of human annotators.

### 4.3 DATASETS

We evaluated the performance of SPE-based attacks and compared them with existing state-of-the-art attacks on four text classification datasets that correspond to various potential applications of adversarial attacks from NLP, such as data augmentation or detection of hateful and offensive speech. Figure 3 shows the average number of words in each of the datasets studied. All source code, links to the datasets, and trained models can be found on Github. We did not include sentence similarity benchmarks such as MRPC (Dolan & Brockett, 2005) because they do not contain the types of sentences that occur in adversarial examples. Therefore, they would not be relevant for our context.

**Offensive tweets.** This dataset is from the TweetEval set of seven multiclass tweet classification tasks (Barbieri et al., 2020) . The tasks are labeled irony, hate, offensive, stance, emoji, emotion, and sentiment. The offensive task consists of classifying tweets as offensive or non-offensive. We used a subset of 1000 sentences from the training split of the dataset for our attacks. This dataset is particularly interesting in the context of adversarial attacks on text because it is based on real harmful content that can be encountered online. Having the ability to correctly recognize and filter these harmful content is a crucial task for many social networks. Adversarial attacks demonstrate potential flaws in the classifiers used to detect this harmful content, but also offer a tool to improve them.

The attacked model is a roBERTa-based (Liu et al., 2019) model trained on 58M tweets and fine-tuned for offensive language identification with the TweetEval benchmark (Barbieri et al., 2020).

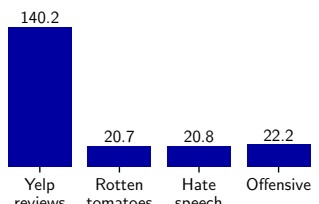

**Hate speech tweets.** This dataset is from the same set of tasks as the Offensive Tweets dataset above (Section 4.3). The goal of the hate speech task is to classify tweets as hate speech or non-hate speech. We used a subset of 1000 sentences from the training split of the dataset for our attack. Similarly to the offensive tweets dataset, this dataset and potential adversarial attacks on related classifiers are particularly useful for the detection of harmful content online.

Figure 3: Average number of words per sentence in each dataset. The Yelp reviews dataset has much longer sentences on average. The two datasets based on Twitter have a built-in character limit (tweet size), and the Rotten tomatoes reviews were pre-processed by the dataset authors to usually contain a single sentence.

The attacked model is a roBERTa-based (Liu et al., 2019) model (same as the model used for the offensive tweets dataset) trained on 58M tweets and fine-tuned for offensive language identification with the TweetEval benchmark (Barbieri et al., 2020).

**Rotten tomatoes.** The rotten tomatoes dataset is a movie review dataset with 5,331 positive and 5,331 negative processed sentences from Rotten Tomatoes movie reviews (Pang & Lee, 2005). We used a subset of 1000 sentences from the training split of the dataset for our attack.

The attacked model is the uncased base DistilBERT (Sanh et al., 2020) model fine-tuned on the rotten tomatoes dataset.

**Yelp reviews.** The Yelp review data set is a data set for binary sentiment classification (Zhang et al., 2015). It contains a set of 560000 highly polar yelp reviews for training and 38000 for testing. It consists of Yelp reviews extracted from the 2015 Yelp Challenge data.

The attacked model is a roBERTa (Liu et al., 2019) model fine-tuned on binary sentiment classifcation from Yelp polarity.

## 5 RESULTS

### (a) Offensive tweets

| Attack | ASR ↑ | rASR ↑ | Time ↓ | Mod. rate |
|---|---|---|---|---|
| **TextFooler+SPE (ours)** | 68.3 | **30.7 (100)** | 1.559 | 16.8 |
| **TextFooler+USE** | 75.4 | 20.4 (100) | 1.749 | 17.4 |
| **TextFooler+BERTScore** | 64.2 | 18.6 (100) | 1.955 | 19.5 |
| **TFAdjusted+SPE (ours)** | 11.7 | **9.1 (100)** | 1.054 | 10.6 |
| **TFAdjusted+USE** | 1.0 | - | 0.865 | 9.1 |
| **TFAdjusted+BERTScore** | 5.4 | 5.4 (47) | 1.167 | 10.4 |

### (b) Hate speech tweets

| ASR ↑ | rASR ↑ | Time ↓ | Mod. rate |
|---|---|---|---|
| 65.6 | **45.9 (100)** | 1.865 | 19.3 |
| 69.1 | 27.0 (100) | 2.111 | 20.9 |
| 65.8 | 37.5 (100) | 2.274 | 20.7 |
| 10.8 | **9.8 (97)** | 1.044 | 12.4 |
| 2.6 | 2.4 (23) | 0.771 | 9.7 |
| 5.7 | 5.7 (51) | 1.187 | 14.3 |

### (c) Rotten tomatoes

| Attack | ASR ↑ | rASR ↑ | Time ↓ | Mod. rate |
|---|---|---|---|---|
| **TextFooler+SPE (ours)** | 89.0 | **61.4 (100)** | 0.746 | 12.8 |
| **TextFooler+USE** | 96.4 | 41.5 (100) | 0.824 | 13.2 |
| **TextFooler+BERTScore** | 92.7 | 38.9 (100) | 0.986 | 13.9 |
| **TFAdjusted+SPE (ours)** | 12.6 | **11.8 (100)** | 0.774 | 12.4 |
| **TFAdjusted+USE** | 0.3 | - | 0.513 | 9.6 |
| **TFAdjusted+BERTScore** | 5.7 | 5.1 (57) | 0.993 | 14.2 |

### (d) Yelp reviews

| ASR ↑ | rASR ↑ | Time ↓ | Mod. rate |
|---|---|---|---|
| 87.4 | **36.7 (100)** | 14.792 | 10.2 |
| 90.5 | 7.2 (100) | 13.874 | 10.6 |
| 89.8 | 7.2 (100) | 16.176 | 11.5 |
| 8.2 | **4.5 (81)** | 36.921 | 10.6 |
| 0.3 | - | 22.325 | 1.2 |
| 2.3 | 1.5 (23) | 32.946 | 5.7 |

Table 2: ASR is the attack success rate (in %), rASR is the estimated real attack success rate obtained from the human survey (in %), the time per sentence is in seconds and the modification rate is the average fraction of words changed per sentence (in %). An upward arrow (↑) indicates that higher is better. All numbers but the estimated rASR were computed on 1000 instances. For the estimated rASR, we report the number of annotated examples in parentheses. For some of the results with TFAdjusted + USE/BERTScore the ASR is so low that few sentences could be submitted for annotation, and the rASR estimates are unconclusive. We omitted rASR values for configurations with less than 10 successful attacks out of 1000 attempts.

The results of our experiments with the six attacks on the four datasets are shown in Table 2. We observe that SPE surpasses other alternatives by a large margin in all configurations, with an estimated real attack success rate (rASR) more than 20% higher than USE or BERTScore with TextFooler and more than 3 percentage points higher with TFAdjusted.

As expected, TextFooler-based attacks have a high ASR, reaching 90% and more while attacking the rotten tomatoes dataset for example. But they also have a steep drop from ASR to rASR; for example, TextFooler+USE has its ASR decrease from 96% to a rASR of 41%, indicating that many of the supposedly successful attacks actually do not have the same meaning according to human annotators. This is due to the weakness of the semantic similarity constraints, as they do not filter out low-quality examples. This can be an issue in practical applications, since the attacked sentence could easily be detected by humans. For all four datasets, this drop is minimal when using SPE as a constraint.

TFAdjusted receives lower ASR scores than TextFooler (no ASR greater than 12%), but the drop between ASR and rASR is low compared to TextFooler, with a maximum drop of 2.4% for TFAdjusted+SPE on the offensive tweets dataset. It shows that the sentences generated by the TFAdjusted attack are of high quality, which is confirmed by the annotators. This is in agreement with the observations of (Morris et al., 2020b) who proposed TFAdjusted to improve the output quality of the TextFooler attack. The quality increase is obtained at the expense of the ASR, which can be considerably lower. For example, TFAdjusted with USE obtains ASR scores lower than 3% on all datasets, making it barely usable as an attack, since it generates less than 25 usable sentences out of a 1000 attacked sentences. The ASR of SPE is higher than the other constraints, which can be interpreted as SPE being less strict. Yet, our encoder still has remarkably low drops from ASR to rASR, and surpasses both other solutions in all examples in terms of rASR. This indicates a better ability to generate successful attacks that maintain the original meaning of sentences.

To give a better understanding of the final adversarial examples and the labels given by human annotators, we included several illustrative examples from the Rotten Tomatoes dataset and the TextFooler attack in Table 3. A more diverse set of examples is available in Appendix Section A.4.

| Original sentence | Perturbed sentence | Average human label (↑) | Encoder |
|---|---|---|---|
| "an impressive if flawed effort that indicates real talent." | "an **wondrous** if flawed effort that indicates real talent." | 3.6 | SPE |
| "an original gem about an obsession with time." | "an original **topaz** about an obsession with **jours**." | 1.6 | |
| "it sounds sick and twisted, but the miracle of shainberg's film is that it truly is romance" | "it sounds sick and **madwoman**, but the miracle of shainberg's **cameraman** is that it **awfully** is romance" | 1.4 | BERTScore |
| "one of the greatest family-oriented, fantasy-adventure movies ever." | "one of the **worst** family-oriented, fantasy-adventure **cinemas** ever." | 1.25 | USE |

Table 3: Adversarial examples produced during our experiments for the Rotten Tomatoes dataset and TextFooler attack with the average human label. An upward arrow (↑) indicates that higher is better. In all of these examples, the semantic similarity metric — which is based on the encoder — found the perturbed sentences to be sufficiently similar to the originals.

## 6    CONCLUSION

Our results show that existing state-of-the-art adversarial attacks produce examples that often do not sufficiently retain meaning and therefore should not be considered successful. We confirmed this by conducting an extensive human survey, showing, for example, that up to 70% of the sentences generated by an attack should be discarded because they do not preserve their original meaning. Although increasing the semantic similarity threshold may improve the quality of examples as (Morris et al., 2020a) claims, it also leads to a lower attack success rate and attacks with limited applicability. The issue lies in the similarity metric used to constrain the generation of sentences in the attacks we tested. Existing similarity metrics are based on models trained in an unsupervised way and thus suffer from several issues, such as antonym disambiguation in the sentence latent space. We tackled this by developing the Semantics-Preserving Encoder, which uses word embeddings extracted from supervised models to improve the semantic similarity metric of adversarial attacks. We obtained a much higher success rate on various datasets with the SPE, and human evaluation showed that the generated sentences can reliably fool humans into thinking that they have the same meaning as the original.

We expect future adversarial attacks to still rely on a semantic similarity metric, which is essential for generating candidate sentences with similar meanings. SPE being usable as a drop-in replacement for these metrics, it is not tied to a particular attack algorithm. SPE can be useful for improving any existing attack and will still be usable with future attacks. While SPE as presented in this paper is based on pre-trained fastText classifiers, it should be straightforward to extend this idea to large pre-trained language models. That should result in even higher accuracy of the attacks, at the cost of greater computational complexity. This kind of extensions could be achieved simply by fine-tuning the language models on classification tasks, where, e.g. sentiment is of high importance. Future work could explore the choice of embeddings used in SPE and the potential improvements that could result from choosing different models and supervised and semi-supervised embedding techniques.

Online communication is still conducted primarily through text. Social networks and messaging platforms must process large amounts of user-generated text to prevent dangerous or harmful content from being spread or targeted at specific individuals. In this context, textual adversarial attacks are

a crucial tool for understanding the shortcomings of existing content filtering models and improving them. Better adversarial attacks mean even better models, and SPE is a step towards building higher-quality attacks, which will continue to highlight issues with our machine learning models and help us train them better.

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

# A    APPENDIX

## A.1    CLASSIFIER SELECTION

There were numerous experiments performed regarding the classifiers selection. The most relevant experiments were performed on the Morris dataset (Morris et al., 2020b). This dataset contains 400 sentence pairs, where each pair is labeled by human annotators whether it preserves the meaning.

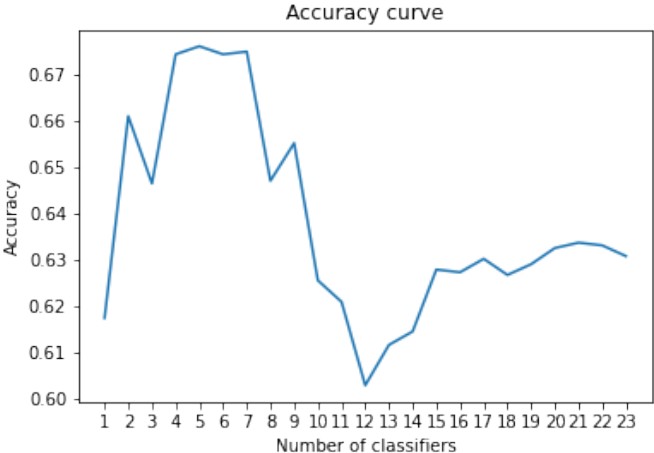

Figure 4: The graph shows how the number of classifiers influences the accuracy of SPE on the Morris dataset (Morris et al., 2020b)

From the Figure 4 we can observe that using just 7 classifiers works the best and then the accuracy drops significantly. These classifiers were trained on Natural Language Inference (NLI) datasets like SNLI (Bowman et al., 2015), cola, rte and sst2 (Wang et al., 2018), and on more downstream tasks such as emotion classification CARER, Yelp reviews and Stack Overflow questions classification (Saravia et al., 2018; Zhang et al., 2015; Annamoradnejad et al., 2022).

We have also experimented with concatenation of embeddings, however empirical results show that averaging works the best.

## A.2 CLASSIFIER TRAINING

In this section, the hyper-parameters of the fastText models (Joulin et al., 2016) that we use in SPE will be specified. FastText models, despite their complexity, are conceptually simple. They employ basic logistic regression and include many hyper-parameters that allow us to adjust the model to our use case (Joulin et al., 2016). Some of the most important hyper-parameters are learning rate, dimensionality of vectors, loss function, and number of epochs. Other parameters such as the minimum and maximum length of char n-grams, word n-grams, number of buckets, minimal and maximal word and label occurrences can be also chosen (Joulin et al., 2016).

In our implementation, most of these hyper-parameters are left with their default value (Joulin et al., 2016). The hyper-parameters which we have found important to modify are:

1. **epoch** - number of epochs
2. **lr** - learning rate
3. **dim** - hidden layer dimensionality (size of the word/sentence vectors)
4. **wordNgrams** - maximum length of word n-grams
5. **minn** - minimum length of char n-gram
6. **maxnn** - maximum length of char n-gram
7. **loss** - loss function used

For the dimensionality of the hidden layer, we have chosen the value 10, which corresponds to the value in the original fastText paper (Joulin et al., 2016). In comparison with the value 512 of USE (Cer et al., 2018), this number is very small, yet we are able to concentrate far more information into just 10 dimensions while achieving SOTA results on classification problems. For the loss function, we use a simple softmax function because our classification problem only has a few classes.

The remaining hyper-parameters were fine-tuned for our use case, which can be done with an automatic tool integrated into the fastText (Joulin et al., 2016) library that finds the best values for the given task for us. The complete list of these hyper-parameters for each classifier is shown in Table 4.

| Classifier | epoch | lr | minn | maxnn | wordNgrams |
|---|---|---|---|---|---|
| SNLI | 5 | 0.05 | 3 | 6 | 4 |
| COLA | 1 | 0.09 | 0 | 0 | 5 |
| RTE | 11 | 0.09 | 6 | 3 | 1 |
| SST2 | 55 | 0.04 | 6 | 3 | 5 |
| StackOverflow | 23 | 0.05 | 6 | 3 | 5 |
| Emotion | 6 | 0.073 | 6 | 2 | 3 |
| Yelp Review Polarity | 5 | 0.05 | 0 | 0 | 2 |

Table 4: Hyper-parameters of the fastText classifiers (Joulin et al., 2016) used in our Semantics-Preserving-Encoder.

Another advantage of the fastText library (Joulin et al., 2016) is the possibility to specify the size of the model. Size compression is achieved using a quantization method, which is very effective. For example, if we take a 409MB classifier with 0.957 accuracy rate in the Amazon Review Polarity dataset (Zhang et al., 2015), we are able to reduce it to 1.5MB while maintaining the same accuracy rate.

Similarly, in our SPE where 7 classifiers are integrated, if each classifier had over 400MB in size, it would make it very difficult to work with the metric that implements it. Quantization solves this issue for us and helps us tremendously. Finally, we decided to limit our model size to 2MB.

## A.3 CLASSIFIER RESULTS

Our metric uses Semantics-Preserving-Encoder, which implements a set of fastText classifiers (Joulin et al., 2016). These were trained on seven datasets with the hyper-parameters stated previously. Each dataset consists of training and testing subsets, which allows us to test the classifiers on the dataset once the training stage is finished. The final accuracy rate obtained on the testing subset of each dataset is presented in Table 5.

| Classifier | Test accuracy rate ↑ |
|---|---|
| SNLI | 0.595 |
| COLA | 0.686 |
| RTE | 0.563 |
| SST2 | 0.829 |
| StackOverflow | 0.891 |
| Emotion | 0.898 |
| Yelp Review Polarity | 0.957 |

Table 5: Accuracy rate of the fastText classifiers used in SPE on the test set for each dataset.

Based on the results in Table 4, we can conclude that better results are achieved on datasets that are more down-stream, such as Emotion, Yelp Review Polarity, or Stack Overflow, where the training was very successful in terms of accuracy.

However, more general tasks such as NLI resulted in a much lower accuracy rate. My explanation is that these problems are very abstract and require a deep language understanding, which we are still unable to reproduce artificially at this point. But overall, even on the less successful tasks, our accuracy is very reasonable. Especially given the fact that we have 3 labels - entailment, contradiction, and neutral, we are still more than two times better than a random classifier.

## A.4 ADVERSARIAL ATTACKS EXAMPLES

We would like to show few examples of textual adversarial attacks with different sentence encoders. Utilizing the information from the human judges, we show one of the best rated and the worst rated ones for each sentence encoder in TextFooler (Jin et al., 2020). Sentence pairs were presented to 5 annotators who assigned them an integer score between 1 and 4, where score 1 strongly disagree, 2 disagree, 3 agree and 4 strongly agree. Only examples from the Rotten Tomatoes dataset (Pang & Lee, 2005) are shown. The Yelp Reviews dataset (Zhang et al., 2015) contains too long pieces of text and hate/offensive tweets dataset (Saravia et al., 2018) is not suitable for the paper due to it's negative nature. However examples of these datasets could be seen in our materials. We released our work as open-source including the human evaluations.

*TextFooler - SPE - Rotten Tomatoes*

| Original sentence | Perturbed sentence | Average human label (↑) |
|---|---|---|
| "for its seriousness , high literary aspirations and stunning acting , the film can only be applauded ." | "for its seriousness , high literary aspirations and stun acting , the film can only be applauded ." | 3.8 |
| "it's one of the saddest films i have ever seen that still manages to be uplifting but not overly sentimental ." | "it's one of the saddest films i have ever seen that still manages to be uplifting but not exceedingly sentimental ." | 3.8 |
| "it sounds sick and twisted , but the miracle of shainberg's film is that it truly is romance" | "it sound sicko and madwoman , but the marvels of shainberg's cameraman is that it exactly is romance" | 1.6 |
| "an original gem about an obsession with time ." | "an original topaz about an obsession with jours ." | 1.6 |

Table 6: Adversarial examples for TextFoolder - SPE - Rotten Tomatoes dataset with the average human label. An upward arrow (↑) indicates that higher is better.

*TextFooler - BERTScore - Rotten Tomatoes*

| Original sentence | Perturbed sentence | Average human label (↑) |
|---|---|---|
| "just another fish-out-of-water story that barely stays afloat ." | "just another fish-out-of-water story that barely maintains afloat." | 3.6 |
| "whether you're moved and love it , or bored or frustrated by the film , you'll still feel something ." | "whether you're moved and love it , or bored or frustrated by the film , you'll anyway feel something ." | 3.4 |
| "has it ever been possible to say that williams has truly inhabited a character ? it is now ." | "has it ever been possible to say that williams gets awfully inhabited a featuring ? it is now ." | 1.4 |
| "elling really is about a couple of crazy guys , and it's therapeutic to laugh along with them ." | "elling altogether is about a couple of lunatic boyfriend , and it's therapeutic to laughing along with them ." | 1.4 |

Table 7: Adversarial examples for TextFoolder - BERTScore - Rotten Tomatoes dataset with the average human label. An upward arrow (↑) indicates that higher is better.

*TextFooler - USE - Rotten Tomatoes*

| Original sentence | Perturbed sentence | Average human label (↑) |
|---|---|---|
| "a worthy entry into a very difficult genre ." | "a reputable entry into a very difficult genre ." | 3.5 |
| "it's a nicely detailed world of pawns , bishops and kings , of wagers in dingy backrooms or pristine forests ." | "it's a mildly detailed world of pawns , bishops and kings , of wagers in dingy backrooms or pristine forests ." | 3.25 |
| "one of the greatest family-oriented , fantasy-adventure movies ever ." | "one of the worst family-oriented , fantasy-adventure cinemas ever ." | 1.25 |
| "an enjoyable film for the family , amusing and cute for both adults and kids ." | "an contented scorsese for the dwelling , goofy and leggy for both grownup and infantile ." | 1.0 |

Table 8: Adversarial examples for TextFoolder - USE - Rotten Tomatoes dataset with the average human label. An upward arrow (↑) indicates that higher is better.

*TFAdjusted - SPE - Rotten Tomatoes*

| Original sentence | Perturbed sentence | Average human label (↑) |
|---|---|---|
| "the story feels more like a serious read , filled with heavy doses of always enticing sayles dialogue ." | "the story feels more like a serious read , filled with heavy dosage of always enticing sayles dialogue ." | 3.8 |
| "lathan and diggs have considerable personal charm , and their screen rapport makes the old story seem new ." | "lathan and diggs have considerable personal glamour , and their screen rapport makes the old story seem new ." | 3.6 |
| "girls gone wild and gone civil again" | "girls gone wild and faded civil again" | 2.6 |
| "has a shambling charm . . . a cheerfully inconsequential diversion ." | "has a shambling charm . . . a blithely trivial diversions ." | 2.2 |

Table 9: Adversarial examples for TFAdjusted - SPE - Rotten Tomatoes dataset with the average human label. An upward arrow (↑) indicates that higher is better.

*TFAdjusted - BERT - Rotten Tomatoes*

| Original sentence | Perturbed sentence | Average human label (↑) |
|---|---|---|
| "the story feels more like a serious read , filled with heavy doses of always enticing sayles dialogue ." | "the story feels more like a serious read , filled with heavy dosages of always enticing sayles dialogue ." | 3.6 |
| "" the emperor's new clothes " begins with a simple plan . . . . well , at least that's the plan ." | "" the emperor's new clothing " start with a simple plans . . . . well , at least that's the plan ." | 3.4 |
| "it's a minor comedy that tries to balance sweetness with coarseness , while it paints a sad picture of the singles scene ." | "it's a marginal comedy that tries to balance sweetness with coarseness , while it painting a sorrowful photo of the singles scene ." | 2.2 |
| "as a director , mr . ratliff wisely rejects the temptation to make fun of his subjects ." | "as a director , mr . ratliff prudently dismiss the temptation to make amusing of his topic ." | 2.2 |

Table 10: Adversarial examples for TFAdjusted - USE - Rotten Tomatoes dataset with the average human label. An upward arrow (↑) indicates that higher is better.

*TFAdjusted - USE - Rotten Tomatoes*

| Original sentence | Perturbed sentence | Average human label (↑) |
|---|---|---|
| "bubba ho-tep is a wonderful film with a bravura lead performance by bruce campbell that doesn't deserve to leave the building until everyone is aware of it ." | "bubba ho-tep is a wondrous film with a bravura lead performance by bruce campbell that doesn't deserve to leaving the building until everybody is aware of it ." | 3.6 |
| "the film has just enough of everything – re-enactments , archival footage , talking-head interviews – and the music is simply sublime ." | "the movie has just enough of everything – re-enactments , archival footage , talking-head interviews – and the music is merely sublime ." | 3.0 |
| "the film starts out as competent but unremarkable . . . and gradually grows into something of considerable power ." | "lthe film starts out as competent but unremarkable . . . and gradually grows into anything of considerable power ." | 3.0 |

Table 11: Adversarial examples for TFAdjusted - USE - Rotten Tomatoes dataset with the average human label. An upward arrow (↑) indicates that higher is better.

