# OpenReview forum: "Preserving Semantics in Textual Adversarial Attacks"
_ICLR.cc/2023/Conference — Submitted to ICLR 2023_

### Official Review · Reviewer_6oWy · 2022-10-20

**Confidence:** 4
**Correctness:** 3
**Technical Novelty And Significance:** 2
**Empirical Novelty And Significance:** 2
**Recommendation:** 3

**Clarity, Quality, Novelty And Reproducibility:**

The paper is clear and easy to follow.


**Strength And Weaknesses:**

**Strength:**

- This paper tackles a critical challenge in textual adversarial attack -- to preserve the semantic meaning of a sentence.
- The paper is mostly clear and easy to follow.


**Weaknesses:**

Several technical decisions are not justified.
- Why use average of embeddings?
- Why use 7 datasets? Does some datasets more helpful then other datasets? For example, what if we remove the semantic classifiers from SPE when attacking Yelp dataset?
- Why use 2.5 as the threshold for rASR? The sentences are just ``slightly similar'' if it is annotated as 3.

There are several issues with the experiments
- Why not verify the method on standard sentence similarity benchmarks such as SemEval? SPE can be considered as a sentence similarity metric. However, this paper neither verifies its performance on standard sentence similarity benchmarks nor justifies that this metric is only suitable for adversarial attack situations.
- Why use distilBERT on Rotten tomato dataset, while using RoBERTa on other datasets?
- There are many other adversarial attack methods, such as BERT-Attack, CLARE, SememePSO, etc. Does SPE also improves the quality for these methods?
- More insights is needed to demonstrate the improvement. For example, certain type of changes are eliminated by applying SPE.

Other questions:
- I'm not sure if Eq. (1) is correct. Why does the complexity depend on the vocabulary size rather then the length of the sentence?
- What do you mean "do not preserve the semantics and even the meaning of the text"?
- What do you mean “eliminate the problem of vocabulary words”?

More philosophical question: Is SPE trying to improve the semantic similarity or to improve prediction consistency?

**Summary Of The Paper:**

Adversarial attack methods on text classifiers fail to preserve semantics of the sentence in some cases, because existing unsupervised sentence encoders fall short in distinguishing synonyms against antonyms. This paper proposes Semantics Preserving Encoder (SPE) which takes the average of multiple sentence embeddings extracted from different text classifiers to address this issue. Experiment results show the proposed method can improve the rASR metric.


**Summary Of The Review:**

This paper proposes SPE to improve the quality of adversarial sentences. The idea is original, but many technical decisions of the method are unjustified, and the experiments need to be significantly improved.

---

> ### Author Response · Authors · 2022-11-15
> **Updated method, experiments and extended appendix**
>
> We thank the reviewer for the insightful comments.
>
> 1. **Technical decision**
>
> Thank you for the valuable inputs. We agree that these points could be better reasoned in our paper. We have extended the method and appendix section and explained the decisions behind these technical details.
>
> 2. **Semantics similarity benchmarks**
>
> We have modified the experiments section to address this issue and make it more clear. We did not include sentence similarity benchmarks such as MRPC because they do not contain the types of sentences that occur in adversarial examples. Therefore, they would not be relevant for our context.
>
> 3.  **distilBERT/RoBERTa**
>
> The choice of attacked model was done for each dataset on the basis of availability of pre-trained fine-tuned models. This does not affect our results since we still compare attacks on a single attacked model+dataset pair at a time in our experiments. Moreover, since these pre-trained models are widely available for each target dataset, the experiments are easier to reproduce and extend without having to fine-tune another large LM.
>
> 4. **Other adversarial attacks**
>
> Even though we would certainly like to evaluate our metric on other attacks, we simply do not have the manpower and resources to do so, especially with regards to the human evaluation and the need to manually label the results. Unfortunately, there is no better way to determine if the attack was successful or not at this moment. However, we believe that the current scope is sufficient as a proof of concept and can always be further explored in the future.
>
> 5. **Insights on more SPE changes**
>
> We respectfully disagree that our results are insufficient to demonstrate the improvement. We clearly demonstrate that we improve the synonym/antonym problem and our solution produces far better semantically correct sentences. Unfortunately, our human evaluation was not aimed to measure the changes, nor to recognise any patterns in the modifications made by SPE. Therefore, we cannot make any claims whether SPE shows improvement in other aspects other than more semantically correct sentences.
>
> 6. **Minor word formulation issues**
>
> We thank the reviewer for highlighting these minor word formulation issues. There was indeed a mistake in the equation. Also, we have addressed the issue with a couple of incorrect formulations or minor typos (semantics/meaning of the text, vocabulary words/out of vocabulary words).
>
> 7. **Philosophical question**
>
> SPE is indeed helping to detect if the semantic in the domain of word level adversarial attacks was preserved.

---

### Official Review · Reviewer_QUPU · 2022-10-23

**Confidence:** 4
**Correctness:** 2
**Technical Novelty And Significance:** 2
**Empirical Novelty And Significance:** 2
**Recommendation:** 5

**Clarity, Quality, Novelty And Reproducibility:**

Clarity: the clarity is good, and the paper is easy to understand.
Quality: the paper has a good motivation that catches a small yet vital problem in adversarial attacks in texts and proposes a very simple method as a solution, yet the evaluation is somewhat unfair which may hurt the quality evolution of the method.
Novelty: the paper catches a problem that is not fully explored in previous works, yet the proposed method is not novel.
Reproducibility: the method can be easily produced.


**Strength And Weaknesses:**

Strength:
A.
the method is very simple and straightforward, aiming to tackle a key problem in substitution-based adversarial sample generation.
The semantic-preserving is indeed an important problem that previous attack methods such as GA, Textfooler, BERT-Attack, PSO-attacks may have trouble solving.
The proposed SPE uses labeled datasets to train an ensemble of classifiers and use all these classification results as the semantic score checking.

B.
The setup of the experiments is reasonable, the human-involved experiment and the real ASR metric are convincing.

Weakness:

A.
My first concern is though the experiments are properly designed, the comparison might not be fair.
That is, as explored thoroughly in prior works such as Textfooler and BERT-Attack, the transferability of the adversarial samples is somewhat weak, that is, the adversarial samples are not universal to different classifiers.
While the proposed SPE is in fact trained with datasets including NLI, cola, rte, sst-2, yelp, etc.,  these datasets can be transferable, therefore, the SPE score can be viewed as another classifier in a way.
I am concerned that the SPE similarity is in fact a reflection of classification, not as illustrated as ‘semantic-preserving’.
If it is the case, then the advantage of low cost in the proposed scorer is less convincing since we can always train some simple classification models as a scorer to measure the adversarial sample quality.

B. semantic preserving concept:
As illustrated in Table 1, the authors intend to illustrate the concept of sentence semantic similarity when the sentence is perturbed.
These cases, which mostly are synonyms to antonyms change, are less observed in the similarity-based word-embedding used in the Textfooler but actually constantly happen in BERT-Attack generated substitutes, which cannot be properly recognized by sentence-level semantic scorers such as USE or BERT-Score.
Therefore, the word-level and sentence-level semantic preserving performances can be different in measuring the quality of adversarial samples yet the paper seems to mix these two concepts.


C. The novelty of the proposed method:
It is true that using USE or BERT-Score in evaluating the quality of the generated adversarial samples is costly and cannot obtain promising results.
Yet the proposed method is somewhat trivial since it only uses an ensemble of classifiers and uses the cosine similarity to calculate the final similarity score calculation.
How can the proposed SPE be used in more general scenarios as a semantic preserving sentence similarity checker is not discussed in the paper.


**Summary Of The Paper:**

This paper introduces a simple yet effective sentence-encoder for the semantic preserving test in the field of word-substitution-based adversarial sample generation.
Generally, the idea is simple since it uses the annotated dataset to train a simple sentence encoder to judge whether the semantics is significantly changed by the attack algorithms.
Compared with previous methods such as USE, the proposed method SPE is simple and efficient.
With a human-involved experiment and a proper metric that measures the real-attack success rate, the proposed method is proven effective to serve as a semantic checking tool in the attack generation process.


**Summary Of The Review:**

This paper is easy to understand and the proposed method is clearly illustrated and simple to implement.
The main concern is fairness in the evaluation process, the authors should provide some analysis of the transferability of attack methods and the connection to their methods.
---------------
I have read the authors' responses and decided to keep my score unchanged. The technical quality of the paper would benefit from a major revision.

---

> ### Author Response · Authors · 2022-11-15
> **Method and experiments updated**
>
> We thank the reviewer for the positive feedback as well as insightful suggestions.
>
> 1. **Experiments**
>
> We agree with the reviewer’s point that the SPE similarity is a reflection of classification. It is thanks to this reflection of classification that we are able to mitigate the problems with antonyms. Unfortunately, it was hard for us to grasp the reason behind the reviewer’s concerns, but we strongly believe our experiments were designed correctly and fairly. We measure the cosine similarity in the vector space for each method, which should be a fair comparison regardless of the specific method. Moreover, we see the potential transferability/universality of SPE and the possibility to use different classifiers to be one of the strengths of our concept and we do not see how this would have any negative impact on the fairness of the results.
>
> 2. **Word level/sentence level**
>
> While we agree that BERTAttack is more inclined to have problems with antonyms, we would like to highlight that these problems are also observable with TextFooler, despite the fact that it uses similarity-based word-embedding. To our knowledge, there is no direct link between the word-level versus sentence-level attacks and the antonyms problem, but we would be happy to discuss and explore this further if there is any research on this topic we may not be familiar with. Also, it is important to mention that both of these attacks use USE for their final decision whether the adversarial example is similar to the original or not. And it often fails to do so, as we discuss in the paper. Therefore, the downsides of USE should be relevant to both attacks.
>
> 3. **Novelty**
>
> Regarding the simplicity of our idea, we consider it to be a strength, rather than a weakness, since we prove it to be successful in resolving or reducing the occurrence of the described problem. This is also supported by the efficiency, universality and potential applicability of our approach to other contexts, where the ease to modify the set of classifiers is an advantage.

---

### Official Review · Reviewer_dmKS · 2022-10-24

**Confidence:** 3
**Correctness:** 2
**Technical Novelty And Significance:** 2
**Empirical Novelty And Significance:** 2
**Recommendation:** 3

**Clarity, Quality, Novelty And Reproducibility:**

The method is clearly presented. The novelty may not be enough. The code is available so I assume the reproducibility is good.

**Strength And Weaknesses:**

Strength

1. The method is clearly presented.

2. The code is available.

Weaknesses

1. I am worried about the novelty. It seems that the proposed method just takes an average of embeddings from many different encoders.

2. The motivation is not clear. The paper found that previous encoders fail to deal well with antonyms, but it seems that the proposed method did nothing special about antonyms either.

3. Some related word-level attacks should be discussed, e.g., [1], [2], [3].

4. Some designs need more justifications. E.g., what is the criterion when choosing the 7 datasets and why the threshold is just set as 2.5.

[1] Moustafa Alzantot, Yash Sharma, Ahmed Elgohary, Bo-Jhang Ho, Mani Srivastava, and Kai-Wei Chang. Generating natural language adversarial examples. In EMNLP, 2018.

[2] Shuhuai Ren, Yihe Deng, Kun He, and Wanxiang Che. Generating natural language adversarial examples through probability weighted word saliency. In ACL, 2019.

[3] Xinshuai Dong, Anh Tuan Luu, Rongrong Ji, and Hong Liu. Towards robustness against natural language word substitutions. In ICLR, 2021.

**Summary Of The Paper:**

This paper proposes a sentence encoder, SPE, to improve the quality of adversarial examples in terms of minimizing semantics change. Specifically, the authors train multiple text classifiers and average the output embeddings from these classifiers to get the final representation of an input sentence. The paper also proposes an rASR metric that employs humans to evaluate how similar an adversarial example is to the vanilla sample.

**Summary Of The Review:**

Given the strength and weaknesses, I tend to reject.

======================After rebuttal==========================

Thank the authors for the effort in answering my questions. After reading all the review comments and responses, I decided to keep my score unchanged.

---

> ### Author Response · Authors · 2022-11-15
> **Related work, method updated and additional experiments added**
>
> We thank the reviewer for the insightful comments.
>
> 1. **Novelty**
>
> While we agree with the statement that our proposed concept is quite simple, we tend to respectfully disagree with the expressed concern about the novelty of our solution. Firstly, our paper explores, highlights and provides evidence for the problem with preserving semantics in adversarial attacks, which to our knowledge is a topic that has only been addressed sparsely. We then propose a solution to this problem, where the novelty lies in the supervised embedding technique, which is something that has not been done in this context before. Also, we consider the simplicity of our idea to be a strength, rather than a weakness, since we prove it to be successful in resolving or reducing the occurrence of the described problem. This is also supported by the efficiency, universality and potential applicability of our approach to other contexts, where the ease to modify the set of classifiers is an advantage.
>
> 2. **Motivation**
>
> We would like to clarify that our proposed method does mitigate the problem with antonyms, which is supported by the results in Table 2, where we can observe a significant improvement in rASR, which corresponds to how well the semantics was preserved in the adversarial attacks based on the human evaluation. From the beginning this was one of the aims of our solution and we approached the problem in the design phase already. We utilise human labelled datasets, where the sentences with an opposite meaning have a different label, thus word embeddings trained on such datasets tend to mitigate the problem with antonyms by design.
>
> 3. **Related work**
>
> We agree with the reviewer’s comment and we have modified the related work section to include other related word-level attacks.
>
> 4. **Design changes**
>
> Since some of these technical design decisions were based on multiple quite complex experiments or factors, we have previously considered this information redundant and did not explain this in depth in the paper. However, thanks to the reviewer’s insight, we admit that this is something that requires reasoning and we have modified the paper to include information on these points (in the method section and partially in the appendix).

---

### Author Response · Authors · 2022-11-15
**Draft Updated**

We thank all the reviewers for their insightful comments and suggestions. We have provided significant updates to our work by incorporating feedback from the reviewers. We summarise the major updates here and encourage all reviewers to check the paper for details.

**Improved method and experiment section** We have also significantly reworked the method and experiments section. More details about the selection of classifiers, thresholds, and additional experiments are discussed in the appendix.
- We clarify the specific choices such as the selection of classifiers, threshold, and the averaging method.
- We explain why it is not beneficial to evaluate our method on sentence similarity benchmarks such as MRPC [5].
- We address other minor mistakes highlighted by the reviewers.

**Improved related work** We have improved related work to incorporate reviewers’ comments.
- More related word-level attacks are included [1, 2, 3, 4]

References:

[1] Moustafa Alzantot, Yash Sharma, Ahmed Elgohary, Bo-Jhang Ho, Mani Srivastava, and Kai-Wei Chang. Generating natural language adversarial examples. In EMNLP, 2018.

[2] Shuhuai Ren, Yihe Deng, Kun He, and Wanxiang Che. Generating natural language adversarial examples through probability weighted word saliency. In ACL, 2019.

[3] Xinshuai Dong, Anh Tuan Luu, Rongrong Ji, and Hong Liu. Towards robustness against natural language word substitutions. In ICLR, 2021.

[4]  Li, D., Zhang, Y., Peng, H., Chen, L., Brockett, C., Sun, M.T., & Dolan, B. (2021). xz. In Proceedings of the Conference of the North American Chapter of the Association for Computational Linguistics.

[5] Dolan, W., & Brockett, C. (2005). Automatically Constructing a Corpus of Sentential Paraphrases. In Proceedings of the Third International Workshop on Paraphrasing (IWP2005).

---

### Decision · Program_Chairs · 2023-01-20

**Decision:**

Reject

**Justification For Why Not Higher Score:**

There are still several remaining concerns after the rebuttal. The paper needs a significant revision before publishing.

**Justification For Why Not Lower Score:**

N/A

**Metareview: Summary, Strengths And Weaknesses:**

The paper presents a sentence encoder technique to preserve semantic information in word-substitution-based adversarial attacks. The proposed approach is straightforward and effective.

Strengthes:

+ The paper is easy to follow and the proposed approach is simple and effective.

+ The experiments, in general, support the main claims.

Weaknesses:

- Although the authors attempt to address the concern of lack of novelty, the problem remains after the rebuttal. I would suggest authors to improve the papers in the following potential directions: 1) position the paper with the literature better to emphasize the novelty of the proposed research; 2) further improve the proposed approach by error analysis; 3) provide a more detailed analysis to justify the design choices. There is no problem with proposing a simple method, but it is essential to motivate the design.

- The overall writing still needs a few rounds of revisions. Although the authors attempt to address some writing issues during the rebuttal, the design choices and experiment details still need to be elaborated.

- The approach should be tested on typical semantic similarity benchmarks (e.g., STS) as the method is based on cosine semantic similarity.

- Another remaining concern is that the classifiers used to build the SPE similarity constraint are highly correlated to the dataset where the attack is conducted. It's unclear if the approach can be generalized to other cases.


**Summary Of Ac-Reviewer Meeting:**

N/A